# Long-Term Effects Following Fresh/Vitrified Embryo Transfer Are Transmitted by Paternal Germline in a Large Size Rabbit Cohort

**DOI:** 10.3390/ani10081272

**Published:** 2020-07-25

**Authors:** Ximo Garcia-Dominguez, José Salvador Vicente, María P. Viudes-de-Castro, Francisco Marco-Jiménez

**Affiliations:** 1Instituto de Ciencia y Tecnología Animal, Universitat Politècnica de València, 46022 Valencia, Spain; ximo.garciadominguez@gmail.com (X.G.-D.); jvicent@dca.upv.es (J.S.V.); 2Centro de Investigación y Tecnología Animal, Instituto Valenciano de Investigaciones Agrarias, 12100 Segorbe, Spain; viudes_mar@gva.es

**Keywords:** assisted reproduction technology, paternal inheritance, maternal inheritance, embryo transfer, embryo cryopreservation, long-term effects, developmental plasticity, developmental programming

## Abstract

**Simple Summary:**

Assisted reproductive technologies (ARTs) involve an extraordinary change in the natural developmental trajectory of the mammalian embryo, incurring potential long-term and inheritable effects in the resulting offspring. The results of this study demonstrate, for the first time, that ex vivo embryo manipulations during fresh and vitrified embryo transfer are associated with paternally inherited bodyweight variation, but seemed not transmissible via the female germline. This asymmetry in the transmission of acquired features following ARTs suggests that embryo paternal and maternal genomes differ in their degree of susceptibility to the lasting effects of ARTs. This study would provide a novel view of developmental plasticity in the early mammalian embryo.

**Abstract:**

The concept of developmental programming suggests that the early life environment influences offspring phenotype in later life, whose effects may also be manifested in further generations. Valuable pieces of evidence come from the fields applying assisted reproductive technologies (ARTs), which deprive embryos of their optimal maternal environment and were thus associated with subsequent developmental deviations. Recently, we demonstrated that the in vitro manipulations during a vitrified embryo transfer procedure incurs a cumulative and transgenerational decline in the growth performance of the resulting offspring. Here, we provide a longitudinal study to investigate whether previous developmental deviations could be indistinctly paternally or maternally transmitted using crossbred mattings. Our findings revealed that early embryo manipulations through fresh and vitrified embryo transfer incurred paternally transmissible effects over the growth pattern and adult body weight, which seemed not inheritable via the female germline. Similar inheritable effects were observed after fresh and vitrified embryo transfer, suggesting that disturbing optimal embryo development through in vitro manipulations was the principal trigger of transmissible effects, rather than embryo cryopreservation per se.

## 1. Introduction

Since the introduction and widespread use of assisted reproductive technologies (ARTs), alterations in embryo viability and quality, together with developmental deviations in the resulting offspring, are starting to emerge [1,2,3]. The underlying hypothesis is that suboptimal in vitro conditions during ARTs may induce stress responses and alterations on developing embryos that can persist in the long term after birth [4,5,6]. In this context, some approaches have been developed in an attempt to mimic in vivo maternal environment, leading to embryos with better quality and improved development [7,8,9]. However, embryo cryopreservation requires embryos to be exposed to environments in which they have no intrinsic ability to survive such as ultralow temperatures and toxic cryoprotectant solutions used to achieve the cryogenic suspension of life [10]. Although the innocuousness of this procedure has already been questioned decades ago, today, the information assessing the long-term effects following the transfer of cryopreserved embryos is very scarce [11]. Recently, we reported a longitudinal study in a rabbit model designed to unravel the effects across the life course following either fresh and vitrified embryo transfer [12]. Although embryo transfer techniques are mandatory in the revitalization of an organism from cryopreserved embryos and cannot be waived from the whole procedure, our findings showed that both ex vivo manipulations during embryo transfer and embryo cryopreservation per se are important cumulative triggers of developmental reshaping and phenotypic deviations that are maintained in adulthood, concordantly with the ARTs literature [1,2,3,13,14,15].

The concept of developmental programming suggests that the early life environment influences offspring characteristics in later life, but growing evidence also suggests that these effects may also be manifested in further generations without further suboptimal exposure [16,17,18,19,20]. Our recent study demonstrated that the transfer of cryopreserved embryos incurs transgenerationally inheritable effects over the offspring growth performance, adult body weight, phenotype of vital organs, and their molecular physiology and metabolism [20]. The developmental plasticity following ARTs is believed to be mediated by epigenetic changes [5,6,21]. However, the epigenetic landscape and remodeling differs both between the sperm and oocyte during gametogenesis and between male and female pronuclei during early embryo development [22,23]. Based on this marked asymmetry in the regulation of paternal and maternal alleles, the plasticity and susceptibility to ARTs of paternal and maternal genomes could differ. Therefore, we hypothesized that the transmission of heritable features after ARTs may be different between the paternally and maternally legacy. In the current study, we performed a longitudinal phenotypic study to evaluate the possible divergence between the maternal and paternal transmission of the growth deviations following fresh and vitrified embryo transfer in a large size rabbit cohort.

## 2. Materials and Methods

### 2.1. Animals and Ethical Statements

New Zealand rabbits belonging to the Universitat Politècnica de València were used throughout the experiment. All animals belonged to a synthetic line selected since the 1980s to increase litter size at weaning [24], with the current 44th generation selected. The animal study protocol was reviewed and approved by the “Universitat Politècnica de València” Ethical Committee (code: 2018/VSC/PEA/0116). All experiments were performed in accordance with the guidelines and regulations set forth in Directive 2010/63/EU EEC and were conducted in an accredited animal care facility (code: ES462500001091).

### 2.2. Experimental Design

Three experimental progenies were previously established to study the long-term effects of the successive ARTs used during a vitrified embryo transfer procedure: one from fresh-transferred (FT) embryos, one from vitrified-transferred (VT) embryos, and another through natural conception (NC) as the control reference [12]. Embryo transfer was performed as described by Besenfelder and Brem [15] and Garcia-Dominguez et al. [25]. Embryo vitrification-warming procedures were adapted from the protocol developed by Vicente and Garcia-Ximénez and described elsewhere using the Cryotop^®^ methodology [12,25,26]. Briefly, 96 fresh and 101 vitrified embryos were transferred into six and seven foster mothers, generating 71 FT and 65 VT animals, respectively. In addition, 73 NC animals were obtained from six pregnant females without any embryonic manipulation. Once the animals reached adulthood, 10 males from each experimental group were used to generate seminal pools and we performed 225 (72 NC, 77 FT, and 76 VT) inseminations in the control females. Likewise, 10 control males were used to generate seminal pools and inseminated 46 (16 NC, 12 FT, and 18 VT) experimental females. Here, our purpose was to carry out a follow-up study of the offspring born after these mattings in order to elucidate whether the long-term effects of the embryo transfer and embryo cryopreservation procedures were transmissible through the male and/or female germ line. To evaluate the paternal transmission, offspring born from experimental males and control females (paternal crossbred animals; Pc) were tracked, while the maternal transmission was monitored in the offspring born from control males and experimental females (maternal crossbred animals; Mc). All rabbit colonies used along this study were housed at the Universitat Politècnica de València animal facilities in flat deck indoor cages (75 × 50 × 40 cm). After weaning, animals were caged collectively (eight rabbits per cage) until the ninth week. Thereafter, animals were individually kept in separate cages. All animals were tracked from birth until adulthood, comparing their growth performance and body weight at birth, weaning (fourth week), prepuberty (ninth week), puberty (fifteenth week), and adulthood (twentieth week).

### 2.3. Statistical Analyses

A general linear model (GLM) was fitted for body weight analysis including the experimental group as a fixed effect and common litter as a random effect. Litter size was used as a covariate for body weight correction, although it remained non-significant from the ninth week of age onward. Growth performance was estimated by nonlinear regression using the Gompertz equation y = *a* exp(−*b* exp(−*k*t)), well suited for rabbits as described before [14,18]. As to the meaning of parameters, *a* can be interpreted as the mature body weight maintained independently of short-term fluctuations; *b* is a timescale parameter related to the initial body weight; and *k* is a parameter related to the rate of maturing (growth rate). Growth [*a* exp(−*b* exp(−*k*t))] and maturity [exp(−*b* exp(−*k*t))] curves were plotted using the estimated parameters. A *p*-value of less than 0.05 was considered indicative of a statistically significant difference. The data are presented as least square mean ± standard error of the mean. All statistical analyses were performed with the SPSS 21.0 software package (SPSS Inc., Chicago, IL, USA).

## 3. Results

Compared to NC animals, both FT and VT animals from the Pc generation exhibited lower body weight at puberty age and adulthood (Table 1). In the Mc generation, FT and VT animals only showed a reduced body weight at weaning time in comparison with NC animals, but these differences were resolved from the age of prepuberty (Table 1).

These results were also supported by the estimated Gompertz parameters (Table 2). Lower a values were noted for the FT and VT progenies of the Pc generation compared with the NC one, that of the Mc generation being similar for all progenies. No differences in the b and k parameters were observed among the experimental groups, either in the Pc or the Mc generation. Thus, growth and maturity curves showed that all the experimental progenies of the Pc and Mc generation exhibited a similar degree of growth and development, but those FT and VT animals of the Pc generation reached a lower mature body weight compared to the NC group (Figure A1).

Overall, the results showed that the long-term effects caused by in vitro embryo manipulations are paternally, but not maternally transmitted to the offspring. Furthermore, no differences were observed between FT and VT animals, suggesting that ex vivo manipulation during the embryo transfer, rather than embryo cryopreservation stressors, are the principal cause of the inheritable effects following vitrified embryo transfer.

## 4. Discussion

Our findings revealed that growth deviations caused following an embryo transfer procedure are paternally transmissible, but did not seem to be inheritable via the female germline. Moreover, we found a comparable growth performance between FT and VT animals from both the Pc and Mc generations, suggesting that disturbing optimal embryo development through its recovery, in vitro handling, and mechanical manipulation during its transfer to a new foster mother was the principal trigger of transmissible effects, rather than embryo cryopreservation per se. Mounting evidence suggests that ARTs incur short- and long-term developmental deviations in the offspring, which are thought to be mediated by epigenetic mechanisms [1,2,3,13,14,15,17,23,27]. Recently, we demonstrated that both mandatory ARTs involved in a vitrified embryo transfer procedure (i.e., embryo transfer and embryo cryopreservation) incurred cumulative developmental effects, reducing the growth performance as in vitro manipulation increased [12]. Moreover, these alterations were transgenerationally propagated [20], which is in line with previous studies [18,19,27,28,29]. Here, we show that modifications in the growth performance is propagated with sex specificity. As a result, gametic transmission of the lower growth performance was propagated to the crossbred generations through the male germline, with indistinguishable effects via the female germline. This finding indicates that gametically programmed information survives to reprogram in the early embryo and persists into adulthood [30], but in a specific way, depending on the parental sex. Lower weaning weight was detected both in the FT- and VT-Mc offspring, but was erased in the long-term. We previously reported that females born following fresh and vitrified embryo transfer exhibited poorer lactation performance [12], which can incur a lower weaning weight in the Mc offspring that seemed to be restored later via compensatory growth, well described in rabbits [31].

In mammals, the maternal and the paternal genome are not functionally equivalent: the sperm and oocyte epigenetic profiles are very different from each other, and the epigenetic remodeling process of both pronuclei after fertilization exhibits allelic differences [22,23]. Considering this asymmetry, it is probable that the paternal and maternal genomes have a different sensitivity to ARTs stressors. Although we recognize that further molecular data are needed to validate this hypothesis, Kohda et al. demonstrated that intracytoplasmic sperm injection differentially impacts the gene expression of paternal and maternal alleles [22,32]. Along these lines, the present study provides evidence of gametic transmission of the ART-induced effects via the paternal germline. These findings emphasize the need to further characterize how ARTs lead to effects that can persist in the course of development and in subsequent generations, in order to better determine their biological relevance.

## 5. Conclusions

Here, we provide valuable information in a rabbit model about how the long-term effects caused during in vitro embryo manipulation can be transmitted to the next generation. In summary, our data revealed that embryo manipulation during fresh and vitrified embryo transfer procedures incur a lower growth performance that was propagated primarily through the male germline.

## Figures and Tables

**Table 1 animals-10-01272-t001:** Bodyweight comparison between naturally conceived (NC), fresh-transferred (FT), and vitrified-transferred (VT) progenies of the paternal crossbred generation and maternal crossbred generation.

Body Weight (g)	Naturally-Conceived	Fresh-Transferred	Vitrified-Transferred
Paternal crossbred Animals (*n*)	(718)	(701)	(748)
Birth	52.7 ± 0.51	53.3 ± 0.62	52.7 ± 0.50
Weaning	559.5 ± 5.59	569.8 ± 7.54	562.7 ± 4.53
Prepuberty	1476.8 ± 18.44	1494.6 ± 16.57	1468.3 ± 14.49
Puberty	3108.1 ± 31.01 ^a^	3011.3 ± 27.87 ^b^	3009.5 ± 30.49 ^b^
Adulthood	3924.2 ± 41.49 ^a^	3758.9 ± 35.75 ^b^	3769.1 ± 38.97 ^b^
Maternal crossbred Animals (*n*)	(136)	(98)	(145)
Birth	55.3 ± 0.93	54.6 ± 1.12	52.9 ± 0.90
Weaning	570.5 ± 10.233 ^a^	514.6 ± 11.05 ^b^	525.6 ± 10.62 ^b^
Prepuberty	1668.7 ± 20.58	1704.4 ± 22.18	1751.8 ± 20.85
Puberty	3124.0 ± 55.76	3162.6 ± 57.21	3133.5 ± 55.76
Adulthood	3655.2 ± 82.99	3707.3 ± 82.99	3628.0 ± 80.88

*n* is the number of animals. ^a,b^ Values with different superscripts within a row differ (*p* < 0.05).

**Table 2 animals-10-01272-t002:** Gompertz parameters of the naturally conceived (NC), fresh-transferred (FT), and vitrified-transferred (VT) progenies of the paternal crossbred generation and maternal crossbred generation.

Gompertz Parameters *	Naturally-Conceived	Fresh-Transferred	Vitrified-Transferred
Paternal crossbred animals			
a	4536.6 ± 42.98 ^a^	4329.4 ± 44.83 ^b^	4349.6 ± 42.50 ^b^
b	4.1 ± 0.04	3.9 ± 0.05	3.9 ± 0.04
k	0.16 ± 0.002	0.16 ± 0.003	0.16 ± 0.003
Maternal crossbred animals			
a	4372.7 ± 75.95	4354.1 ± 109.71	4157.6 ± 94.133
b	3.8 ± 0.07	4.0 ± 0.14	3.9 ± 0.13
k	0.16 ± 0.004	0.16 ± 0.007	0.17 ± 0.007

* Gompertz parameters: a can be interpreted as the mature body weight maintained independently of short-term fluctuations; b is a timescale parameter related to the initial body weight; and k is a parameter related to the rate of maturing (growth rate). ^a,b^ Values with different superscripts within a row differ (*p* < 0.05).

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
