# Peer review of "Long-Term Effects Following Fresh/Vitrified Embryo Transfer Are Transmitted by Paternal Germline in a Large Size Rabbit Cohort"

_animals, 2020, doi:10.3390/ani10081272_

Round 1
Reviewer 1 Report
The work by Garcia-Dominguez et al presented a finding of significant interest to the ART community, taking advantage of their unique model colony – rabbits. I have two main comments.
(1)Please change the title to “….in a large size rabbit cohort”. One cannot draw a conclusion to support the title statement in a species yet, based on the data presented.
(2) Any pedigree information on animals of different groups? Any possibility that the body weight differences are due to different pedigrees? It’s not uncommon to see body difference among two colonies of rabbits. Authors should provide information and discussion.
The work by Garcia-Dominguez et al aimed to seek answer to a significant question on whether/how epigenetic marks are transmitted, in the context of assisted reproduction technology (ART). The well-established ART platform in rabbits by the team and the unique reproductive “quickness and easiness” of this species allowed authors to conduct this long term and relatively large-scale study. The findings are straightforward yet powerful, as stated in the title “Long-term effects following fresh/vitrified embryo transfer are transmitted by paternal germline in rabbits”.
That said, the presented data cannot support the conclusion at a species level yet. Please change the title to “Transgenerational effects following fresh/vitrified embryo transfer are transmitted by paternal germline in a large size rabbit cohort”.
This is the first report, to the reviewer’s knowledge, that is conducted in a non-murine species. The authors had a focus on the effects of cryopreservation, which is not systematically studied yet. The findings are novel and striking.
As stated earlier, please change the title to “Transgenerational effects following fresh/vitrified embryo transfer are transmitted by paternal germline in a large size rabbit cohort” to avoid overstate the findings.
I like the work as is. There is no need to expand the paper by adding non-essential evidence. The work is simple but clear and powerful and provides novel information to the research community.
Although the authors suggest the roles of epigenetic marks (which may very likely be the case), there are no molecule evidence in this work to support the involvement of epigenetics. The authors should acknowledge this in the second paragraph of Discussion.
Author Response
The work by Garcia-Dominguez et al presented a finding of significant interest to the ART community, taking advantage of their unique model colony – rabbits. I have two main comments.
(1) Please change the title to “….in a large size rabbit cohort”. One cannot draw a conclusion to support the title statement in a species yet, based on the data presented.
We are entirely in agreement with this suggestion, and we implemented this title to provide an accurate meaning of the manuscript content.
(2) Any pedigree information on animals of different groups? Any possibility that the body weight differences are due to different pedigrees? It’s not uncommon to see body difference among two colonies of rabbits. Authors should provide information and discussion.
Yes, all animals belonged from a synthetic line selected by the Universitat Politècnica de València since the 80s to increase litter size at weaning, being the current generation of selection the 44th. This information and the appropriate reference has been implemented in the manuscript. This synthetic rabbit line is well characterized, and all of the colonies used in this study were animal facilities in the same facilities to reduce confounding factors.
The work by Garcia-Dominguez et al aimed to seek answer to a significant question on whether/how epigenetic marks are transmitted, in the context of assisted reproduction technology (ART). The well-established ART platform in rabbits by the team and the unique reproductive “quickness and easiness” of this species allowed authors to conduct this long term and relatively large-scale study. The findings are straightforward yet powerful, as stated in the title “Long-term effects following fresh/vitrified embryo transfer are transmitted by paternal germline in rabbits”. That said, the presented data cannot support the conclusion at a species level yet. Please change the title to “Transgenerational effects following fresh/vitrified embryo transfer are transmitted by paternal germline in a large size rabbit cohort”. This is the first report, to the reviewer’s knowledge, that is conducted in a non-murine species. The authors had a focus on the effects of cryopreservation, which is not systematically studied yet. The findings are novel and striking. As stated earlier, please change the title to “Transgenerational effects following fresh/vitrified embryo transfer are transmitted by paternal germline in a large size rabbit cohort” to avoid overstate the findings.
I like the work as is. There is no need to expand the paper by adding non-essential evidence. The work is simple but clear and powerful and provides novel information to the research community.
Although the authors suggest the roles of epigenetic marks (which may very likely be the case), there are no molecule evidence in this work to support the involvement of epigenetics. The authors should acknowledge this in the second paragraph of Discussion.
We fully understand that further molecular data will be needed to validate if the asymmetric epigenetic landscape between paternal and maternal information is the real cause of the differences noted between the paternal and maternal crossbred generations. The difficulty is in obtaining funds to undertake this analysis. As we have commented before, with the phenotypic pieces of evidence observed in the present study, we are going to try to obtain the appropriate financial support for the analysis at the epigenomic level.
Reviewer 2 Report
General comments:
The authors performed that in a rabbit model, that the transfer of both fresh and vitrified embryos incurs a cumulative decline in the growth performance of the resulting offspring. They argued that the outcomes of this study provide a valuable reference to better assess the epigenetic landscape of the paternal and maternal genome in early mammalian embryos, regarding its plasticity and the expression dynamics of secondary ART stressors. The review thinks that although the finding is interesting, the MS is more based speculation on abnormal epigenetic regulation, not on the results of aberrant epigenetic modification due to in vitro manipulation. There are many epigenetic aspects required to investigation, such as H19, XCI imprinting, epigenetic landscape of DNA methylation, histone K27, K4 and K27 methylation pattern etc, if the authors try to relate the low growth rate with epigenetic problem. On the other hand, IVF embryos and frozen-thawed embryos experienced large offspring syndrome (LOS) instead of lower body weight, the birth weight should be included. Meanwhile, maternal cross-bred animals resulted in a low Weaning weight and paternal cross-bred animals had lower puberty and adulthood weight, it is difficult to conclude why maternal effect is not the case while paternal influence is. This review suggests that the growth is controlled by gene(s) traits, it is reasonable to argue that this finding may be related to dysregulation of gene interaction of growth related genes, or epigenetic disturbing, rather that arguing a broad and global epigenetic landscape. The argument may be true from the authors, but without sufficient experimental results.
Other Comments:
- Remove any terms of “the first time” etc. through the MS, it is novel and the first, of course, if the MS is subject to be publish.
Author Response
The authors performed that in a rabbit model, that the transfer of both fresh and vitrified embryos incurs a cumulative decline in the growth performance of the resulting offspring. They argued that the outcomes of this study provide a valuable reference to better assess the epigenetic landscape of the paternal and maternal genome in early mammalian embryos, regarding its plasticity and the expression dynamics of secondary ART stressors. The review thinks that although the finding is interesting, the MS is more based speculation on abnormal epigenetic regulation, not on the results of aberrant epigenetic modification due to in vitro manipulation.
Effectively, once the comments of both reviewers have been revised carefully. We recognise and agree that the manuscript states an excess of speculation about the epigenetic mechanisms. We acknowledge that it could be a possible explanation for the outcomes of this study, but further molecular data is required to validate this hypothesis. Over-speculation has been eliminated. Besides, we recognise in the discussion that further molecular evidence is needed to confirm this hypothesis.
There are many epigenetic aspects required to investigation, such as H19, XCI imprinting, epigenetic landscape of DNA methylation, histone K27, K4 and K27 methylation pattern etc, if the authors try to relate the low growth rate with epigenetic problem.
We recognise that the manuscript has a lot of speculations aimed to explain the outcomes of this study. We have focused this reviser version on the transgenerational effects associated with assisted reproduction techniques based on our large size phenotypic cohort. We have reduced all of the comments related with the epigenetics to a couple of simple statements to open a gate towards a new approach that could validate the molecular basis for the phenotypic evidence noted along with this work. We would love to be able to carry out these studies, but currently, we do not have the funding. However, we think that the data presented here can be relevant, given the increase that assisted reproduction techniques and their long-term and transgenerational effects are having today.
On the other hand, IVF embryos and frozen-thawed embryos experienced large offspring syndrome (LOS) instead of lower body weight, the birth weight should be included.
We are entirely in agreement. Initially, we don’t include this information as the “b” parameter of the Gompertz equation are a timescale parameter related to the initial body weight without any short-term influence, and no difference was noted between the experimental groups. However, we note that, as the reviewer indicated, it is interesting to include the birth weight value of the studied progenies in light of the current background provided by the scientific literature.
Meanwhile, maternal cross-bred animals resulted in a low Weaning weight and paternal cross-bred animals had lower puberty and adulthood weight, it is difficult to conclude why maternal effect is not the case while paternal influence is.
We appreciate this comment. It is difficult to affirm if the outcomes noted may be due to inheritable molecular changes without the appropriate molecular data. However, we understand that one way to resolve this issue is by looking at the long-term phenotype (as indicated in the title) since rabbits can resolve any effect in the short and medium-term through compensatory growth (until 45% of the body weight during growth rate). We have evidence that both fresh and vitrified embryo-derived females show more reduced lactation performance (appropriate reference is provided), we consider it more reasonable to explain the weaning weight differences, rather than inheritable molecular changes, as weaning differences are long-term resolved.
This review suggests that the growth is controlled by gene(s) traits, it is reasonable to argue that this finding may be related to dysregulation of gene interaction of growth related genes, or epigenetic disturbing, rather that arguing a broad and global epigenetic landscape. The argument may be true from the authors, but without sufficient experimental results.
Taking into account the comments of both reviewers, we have noticed that our previous version displayed an excess of speculation about the epigenetic changes that, although they may partially explain our results, are not the subject of this study. Thus, we understand the confusing thinking can be induced in the reader since there is no evidence at the molecular level to support this theory. In the present version, we fixed our discussion on the asymmetry in the transmission of acquired features following ARTs in a large size rabbit cohort. The epigenetics speculations have been reduced to a twosome of comments to leave the opening for future studies to verify our results.
Round 2
Reviewer 2 Report
Comments are resolved.